# The Relationship of *ABCB1/MDR1* and *CYP1A1* Variants with the Risk of Disease Development and Shortening of Overall Survival in Patients with Multiple Myeloma

**DOI:** 10.3390/jcm10225276

**Published:** 2021-11-13

**Authors:** Szymon Zmorzynski, Magdalena Wojcierowska-Litwin, Sylwia Popek-Marciniec, Aneta Szudy-Szczyrek, Wojciech Styk, Sylwia Chocholska, Agata Anna Filip

**Affiliations:** 1Department of Cancer Genetics with Cytogenetic Laboratory, Medical University of Lublin, 20-059 Lublin, Poland; magdalena.wojcierowska-litwin@umlub.pl (M.W.-L.); sylwiapopek@umlub.pl (S.P.-M.); agata.filip@umlub.pl (A.A.F.); 2Chair and Department of Hematooncology and Bone Marrow Transplantation, Medical University of Lublin, 20-059 Lublin, Poland; aneta.szudy-szczyrek@umlub.pl (A.S.-S.); sylwia.chocholska@umlub.pl (S.C.); 3Department of Psychology, Institute of Pedagogy and Psychology, Warsaw Management University, 03-772 Warsaw, Poland; wojciech.styk@gmail.com

**Keywords:** multiple myeloma, genetic variants, polymorphisms, *ABCB1* gene, *CYP1A1* gene

## Abstract

(1) Background: The aim of our study was to analyze the possible relationship of *ABCB1* and *CYP1A1* gene variants with susceptibility and outcome of multiple myeloma (MM); (2) Methods: Genomic DNA samples from 110 newly-diagnosed MM patients and 100 healthy blood donors were analyzed by methods-PCR-RFLP (for *ABCB1* 3435C > T, *CYP1A1* 6235T > C—m1), automated DNA sequencing (for *ABCB1* 1236C > T, 2677G > T/A) and allele-specific PCR (for *CYP1A1* 4889A > G—m2); (3) Results: The genotypic frequencies of *CYP1A1* 4889A > G variant were not in Hardy-Weinberg equilibrium for MM patients. The presence of m1 and m2 *CYP1A1* alleles decreased the risk of MM—OR = 0.49 (*p* = 0.011) and OR = 0.27 (*p* = 0.0003), respectively. In turn, TT genotype (*ABCB1* 2677G > T/A) increased the risk of this disease (*p* = 0.007). In the multivariate Cox analysis CT + TT genotypes (*ABCB1* 3435C > T) were associated with decreased risk of death (HR = 0.29, *p* = 0.04). In log-rank test in patients with CT genotype (*ABCB1* 3435C > T) was observed association of overall survival with the type of treatment; (4) Conclusions: Our findings suggest that T-alleles of *ABCB1* 2677G > T/A and m1/m2 alleles of *CYP1A1* affected the susceptibility of MM. Moreover, T-allele of *ABCB1* 3435C > T might be independent positive prognostic factor in MM.

## 1. Introduction

Multiple myeloma (MM) is a malignant disease characterized by clonal expansion of plasma cells in the bone marrow [1]. Multi-drug resistance (MDR) is one of the major problems, which occurs during clinical therapy [2]. Among many mechanisms of MDR, the most prominent role is played by multidrug resistance-associated protein-1 (MRP1), also known as permeability-glycoprotein (P-GP) or ATP-binding cassette sub-family B member 1 (ABCB1) [3]. It acts as a transmembrane efflux pump that transfers drugs and toxins from cytosol to the extracellular matrix [4]. The ABCB1 protein is involved in the body’s defense against endogenous and exogenous toxic compounds [5]. It is encoded by *ABCB1* gene also known as *MDR1* or *P-GP* (*locus* 7q21). A number of single nucleotide variants were identified in the *ABCB1* gene, of which the most common studied are variants 3435C > T (in exon 26, rs1045642), 1236C > T (in exon 12, rs1128503) and 2677G > T/A (in exon 21, rs2032582) [6,7]. These variants may affect protein expression [8]. The variant 2677G > T/A affects the protein sequence (p.Ala893Ser/Thr) and function [6]. In contrast, variants present at nucleotides 3435 [ATC > ATT] and 1236 [GGC > GGT] are caused by synonymous substitution—Ile1145Ile and Gly412Gly, respectively [8]. The presence of the T-allele of the 3435 variant is associated with lower gene expression compared to C-allele [9]. The variants 3435C > T, 1236C > T and 2677G > T/A could impact the transport ability of P-gp-mediated substrates and might be in linkage with other variants associated with MM predisposition [10,11].

ABCB-1-associated drug resistance is observed in about 75% of MM patients treated with proteasome inhibitors [12]. Although MM treatment is usually not curative, the introduction of proteasome inhibitors and new immunomodulatory drugs resulted in improvement of overall survival (OS) relative to previous observations [13,14,15]. 

Bortezomib, as a proteasome inhibitor, induces an apoptotic cascade and has improved clinical outcomes [16,17]. In bortezomib-resistant MM cells the upregulation of the ABCB1 protein was observed at various levels [18]. However, conflicting data exist for bortezomib as a substrate for ABCB1 [19,20]. Some patients do not respond to bortezomib treatment, or relapse after a response is observed [21,22]. To overcome MDR, the combinations of several drugs, including thalidomide as an immunomodulatory drug, are used [17]. The correlations between response to treatment and *ABCB1* gene variants have been not thoroughly studied in MM.

Cytochromes P450 (CYP) are enzymes that play an important role in the metabolism of a wide range of endogenous and foreign compounds [23]. Carcinogen-metabolizing proteins might be involved in individual susceptibility to cancers [24]. The variants of their genes, by altering protein function, might influence the carcinogen activation or deactivation and modulate DNA repair mechanisms [25]. Cytochrome P450 1A1 (CYP1A1) belongs to one of the most active isoenzyme subfamilies and participates in the metabolism of xenobiotics and anti-cancer drugs [26]. The CYP1A1 enzyme can accelerate bortezomib metabolism, leading to its concentration reduction in MM cells, and is encoded by the *CYP1A1* gene (*locus* 15q22q24) [27]. CYP1A1 causes oxidative deboronation of bortezomib, which is the first step of this drug metabolism [28]. The inhibition of the CYP1A1 enzyme increases bortezomib sensitivity in MM cells [27]. *CYP1A1* gene variants have been linked to cancer susceptibility, for example in acute myeloid leukemia, prostate cancer, larynx cancer, lung cancer and MM [25,29,30,31,32]. Genetic variants of the *CYP1A1* gene are present in exons and non-coding regions. In the present article nomenclature system for *CYP1A1* alleles recommended in https://www.pharmvar.org/gene/CYP1A1 (accessed on 30 April 2021) has been followed. In the 3′-UTR at position g.11229T > C (c.6235/c.3798) variant (rs4646903) is located, containing a single thymine (5′-GGGTCC-3′, wild type allele, WT) to cytosine (5′-GGGCCC-3′, genetic variant m1, *CYP1A1*2A*). *CYP1A1* 6235T > C variant is also known as MspI polymorphism as it gains the restriction site for this enzyme [33]. Moreover, it can increase the expression of the gene or the stability of messenger RNA, which results in higher enzyme activity [34]. The second genetic variant (rs1048943) is caused by substitution of adenine (wild type allele, WT) to guanine (m2 allele, *CYP1A1*2C*) at position g.9885A > G (c.4889/c.2454) in exon 7 (ATT to GTT) [31]. The presence of valine (p.Ile462Val) in amino acid chain seems to increase the enzyme activity [35]. Both variants are associated with enhanced metabolism of corresponding substrates and may play an important role in an individual susceptibility to cancer development, including MM [31].

Taking the above into account, we hypothesize that the *ABCB1* and *CYP1A1* variants might be associated with MM risk development, as well as response to treatment with thalidomide and/or bortezomib. We decided to analyze these two genes together, because ABCB1 may transport bortezomib to the extracellular matrix, and the CYP1A1 enzyme is involved in the bortezomib metabolism. Upregulation of ABCB1 and the higher activity of CYP1A1 may lead to lower sensitivity to bortezomib. Although the research of *ABCB1* (mostly rs1045642) and *CYP1A1*2A* variants was made in MM, in our research we expanded the examined factors to include smoking, type of treatment and response to treatment [6,30]. Furthermore, we have checked whether these variants predict sensitivity to bortezomib in cell cultures derived from studied patients.

## 2. Materials and Methods

### 2.1. Patients and Samples

The study group consisted of 110 newly-diagnosed patients with MM (Caucasian population), who were hospitalized (in years 2013–2019) at the Chair and Department of Hematooncology and Bone Marrow Transplantation, Medical University of Lublin. The study received a positive review (no. KE-0254/165/2013 and no. KE-0254/337/2016) from the Bioethics Committee of Medical University of Lublin, according with the ethical standards established by Helsinki Declaration. Detailed patients characteristics are shown in Table 1.

The control samples (Caucasian population) were obtained from 100 healthy blood donors (50 males and 50 females, with mean age 34.4 years) attending the regional Blood Donation and Blood Treatment Center in Kielce, Poland. All participants of the study provided written informed consent.

We followed the methods of Zmorzynski et al. 2020 [36]. The therapeutic induction regimens consisted of thalidomide and/or bortezomib combined with steroids and/or cyclophosphamide. Response to treatment was evaluated according to the International Myeloma Working Group guidelines, as described elsewhere [37,38]. Overall survival (OS) encompassed time from diagnosis until relapse, progression, death due to tumor effect or last follow-up, and time from diagnosis until death by any cause or last follow-up, respectively. Progression free survival (PFS) was estimated as the time elapsed between treatment initiation and tumor progression or death from any cause [36,39].

Peripheral blood (from healthy blood donors) and bone marrow aspirates (from MM patients) were used for DNA isolation and the determination of the *ABCB1* and *CYP1A1* variants.

Cell cultures from bone marrow aspirates were established to carry out the research associated with bortezomib (*n* = 50), as described by Zmorzynski et al. 2020 [36].

### 2.2. DNA Isolation

DNA was isolated with the use of a commercial kit (Qiagen, Hilden, Germany) according to manufacturer’s procedure. The concentration and quality of DNA was checked using NanoDrop device (Thermo Fisher Scientific, Waltham, MA, USA).

### 2.3. ABCB1 Genotyping

The *ABCB1* variants 1236C > T, 2677G > T/A were analyzed using automated DNA sequencing. The genotyping of 3435C > T variant was performed by polymerase chain reaction restriction fragment length polymorphism (PCR-RFLP).

#### 2.3.1. Automated DNA Sequencing

The following primers were used to amplify 346 bp (containing 1236 nucleotide) and 223 bp (containing 2677 nucleotide), respectively.

For 1236C > T:forward 5′-TCA GTT ACC CAT CTC GAA AAG AA-3′reverse 5′-ACA TCA GAA AGA TGT GCA ATG TG-3′

For 2677G > T/A:forward 5′-TAT GGT TGG CAA CTA ACA CT-3′reverse 5′-CAT GAA AAA GAT TGC TTT GA-3′

Each PCR mixture (25 µL) contained 100 ng (for 1236C > T)/50 ng (for 2677G > T/A) genomic DNA, PCR buffer (Clontech, Takara Bio USA, San Jose, CA, USA) (for 1236C > T)/PCR buffer (A&A Biotechnology, Gdańsk, Poland) (for 2677G > T/A), dNTPs mixture (0.25 mM), HD polymerase (0.31 U) (Clontech, Takara Bio USA, San Jose, CA, USA) (for 1236C > T)/RUN polymersase (0.25 U) (A&A Biotechnology, Gdańsk, Poland) (for 2677G > T/A) and primers (10 µM of each). The mixture was heated for 94 °C for 5 min and underwent 35 cycles of amplification: denaturation 98 °C for 10 s (for 1236C > T)/94 °C for 30 s (for 2677G > T/A), annealing 56 °C for 10 s (for 1236C > T)/59 °C for 30 s (for 2677G > T/A), elongation 72 °C for 20 s (for 1236C > T)/72 °C for 40 s (for 2677G > T/A). The final elongation taken 4 min at 72 °C. The PCR reactions were performed in a Applied Biosystems 9700 Thermal Cycler (Applied Biosytems, Waltham, WA, USA). Sequencing PCR was performed using BigDye Terminator v3.1 Cycle Sequencing Kit (Applied Biosystems, Waltham, MA, USA) in thermal cycler (as previously). The sequencing PCR product was purified by use of exterminator kit (A&A Biotechnology, Gdańsk, Poland). The sequencing run module was StdSeq50_POP7 in genetic analyzer 3130 (Applied Biosytems, Waltham, MA, USA). The results were analyzed using Applied Biosystems software (Figure 1).

#### 2.3.2. PCR-RFLP

For analysis of 3435C > T variant PCR-RFLP method was applied according to validated protocol of Turgut et al., 2006 [40]. *ABCB1* gene fragment length of 238 bp was amplified by PCR using primers:-forward 5′-TGC TGG TCC TGA AGT TGA TCT GTG AAC-3′-reverse 5′-ACA TTA GGC AGT GAC TCG ATG AAG GCA-3′

Each PCR mixture (25 µL) contained 100 ng genomic DNA and PCR buffer (Clontech, Takara Bio USA, San Jose, CA, USA), dNTPs mixture (0.25 mM), HD polymerase (Clontech, Takara Bio USA, San Jose, CA, USA) and primers (10 µM of each). The mixture was heated 94 °C for 4 min and underwent 35 cycles of amplification: denaturation 98 °C for 15 s, annealing 60 °C for 15 s, elongation 72 °C for 20 s. The PCR reaction was performed in a Applied Biosystems 9700 Thermal Cycler (Applied Biosytems, Waltham, MA, USA). 

The PCR product was digested with *MboI* (Thermo Fisher Scientific, Waltham, MA, USA) according to manufacturer’s procedure. Restriction enzyme produced two fragments of 172 bp and 60 bp or one fragment of 238 bp for presence of C or T allele, respectively. The fragments were analyzed on 3% agarose gel, stained with SimplySafe (Eurx, Gdańsk, Poland) and visualized in G:Box (Syngene, Cambridge, UK) (Figure 2A). For each sample was carried out an independent PCR analysis.

### 2.4. CYP1A1 Genotyping

For analysis of *CYP1A1* 6235T > C and 4889A > G variants PCR-RFLP and allele-specific PCR methods were used according to protocols of Kumar et al., 2010 and Drakoulis et al., 1994, respectively [41,42]. 

For 6235T > C a total of 25 μL reaction mixture consisted of 100 ng genomic DNA, 10µM of each primer, 0.25 mM dNTPs mixture and 0.31 U of HD polymerase (Clontech, Takara Bio USA, San Jose, CA, USA) with 1× PCR reaction buffer (Clontech, Takara Bio USA, San Jose, CA, USA). The primers for PCR reaction were following:-forward 5′-AAG AGG TGT AGC CGC TGC ACT-3′-reverse 5′-TAG GAG TCT TGT CTC ATG CCT-3′

The thermal profile consisted of initial denaturation 95 °C for 5 min followed by 35 cycles of 98 °C for 15 s, annealing 65 °C for 15 s, elongation 72 °C for 25 s. The PCR reaction was performed in a Applied Biosystems 9700 Thermal Cycler. The restriction enzyme MspI (Thermo Fisher Scientific, Waltham, MA, USA) was used to distinguish 6235T > C variant. The wild type allele (WT, *CYP1A1*1*) produced single band representing fragment of 340 bp, and the variant allele (m1, *CYP1A1*2A*) resulted in two fragments of 200 bp and 140 bp. The heterozygote produced all three bands (Figure 2B).

For analysis of 4889A > G variant was made amplification in 25 μL and reaction mixture was as follows—100 ng genomic DNA and PCR buffer (A&A Biotechnology, Gdańsk, Poland), dNTPs mixture (0.25 mM), RUN polymerase (0.25 U) (A&A Biotechnology, Gdańsk, Poland) and primers (10 µM of each). The mixture was heated 95 °C for 5 min and underwent 35 cycles of amplification: denaturation 95 °C for 20 s, annealing 65 °C for 20 s, elongation 72 °C for 30 s. In reaction were used primers producing 1472 bp:-forward 5′-ATA GGG TTA GTG GGA GGG ACA CG-3′-reverse 5′-GCT CAA TGC AGG CTA GAA CTA GAA TAG AAG-3′

PCR product was divided into two parts (10 μL each, called P57 and P58) and amplified fragments were reamplified. The reaction mixture was the same as that for the first step except the primer mixture—reverse primer was used in two separate reactions. In the first reaction (so called P57), wild type primer (P57, WT, *CYP1A1*1*) 5′-GAA GTG TAT CGG TGA GAC CG -3′ was used. The second reaction was carried out with the variant primer (P58, m2, *CYP1A1*2C*) 5′-GAA GTG TAT CGG TGA GAC CG-3′. The mixture was heated 95 °C for 5 min and underwent 20 cycles of amplification—touchdown PCR: denaturation 95 °C for 30 s, annealing 66 °C (−0.5 °C per cycle) for 20 s, elongation 72 °C for 30 s; and 19 cycles with an annealing temperature of 56 °C for 30 s (other conditions remain the same as in touchdown PCR). The obtained product of 996 bp was visualized in G:Box (Syngene, Cambridge, UK). The presence of strong P57 band indicated a wild type allele, while the presence of a strong P58 band indicated a variant allele. Both bands were observed in the case of heterozygtes (Figure 2C).

### 2.5. Cytogenetic Analyses

Cytogenetic abnormalities observed in MM patients, such as del(17p13.1) and *IgHV* gene rearrangements—t(4;14), t(14;16) were analyzed by simultaneous staining of cytoplasmic immunoglobulin and FISH (cIg-FISH) according to Ross et al. 2012 recommendations and with the use of previously described protocol with modifications [43,44,45]. 

### 2.6. Bortezomib In Vitro Treatment

Bone marrow aspirates (*n* = 50) (mean number of plasma cells—31.31% ± 20.69) were used with established cell cultures as described by Zmorzynski et al. 2019 [46]. Bortezomib (LC Laboratories, Woburn, MA, USA, 200 mg/mL) was dissolved in DMSO and added to cell cultures. The final concentration of DMSO in culture medium was lower than 0.1%. The cell cultures without bortezomib were used as a control. The cultures were grown without G-CSF and were routinely terminated. To determine the number of apoptotic, necrotic and viable cells Annexin V-Cy3 Apoptosis Detection Kit (Sigma-Aldrich, St. Louis, MO, USA), as well as fluorescence microscope were used. The viable cells were stained with 6-CF (6-carboxyfluorescein)—green dye, and necrotic cells showed red staining due to the presence of AnnCy3 (Annexin V Cy3.18). Cells starting apoptotic process were stained with green and red dyes. Plasma cells were analyzed according to Carter’s et al. [47].

### 2.7. Statistical Analysis

Laboratory/clinical values were compared with studied genetic variants using an independent t-test and Chi-square test for continuous variables and categorical variables, respectively. The association of analyzed variants with clinical data was evaluated using Chi-square test or Fisher’s exact test. The quantitative data was shown as frequency or percentage. Deviation of genotype frequencies in controls (healthy blood donors) and cases (MM patients) from Hardy-Weinberg equilibrium (HWE) was assessed by Chi-squared test [48]. The Cox proportional hazard model was used for univariate and multivariate analysis of OS and PFS. The Kaplan-Meier method and the log-rank test were used for survival analysis. We assumed a 5% error of inference and a related level of significance *p* < 0.05 pointing to the existence of statistically significant differences. Statistical analyzes were performed using the Statistica ver. 12.5 (StatSoft) software.

## 3. Results

### 3.1. Frequencies of Alleles and Genotypes

Baseline characteristics for 110 MM patients included in this analysis are shown in Table 1. Genotyping of studied genetic variants was successfully performed for MM patients and healthy blood donors. The HWE test showed that the genotypic frequencies of *CYP1A1* 4889A > G variant were not in HWE for MM patients, because the *p* value was lower than 0.05 and χ^2^ was higher than 3.84 (Table 2).

### 3.2. Studied Variants and the Risk of MM Development

The *CYP1A1* 6235T > C and *ABCB1* gene variants were balanced. The differences in genotypic and allelic frequencies of *ABCB1* 2677G > T/A (TT genotype and T allele), as well as *CYP1A1* variants—6235T > C (WT/m1 genotype and m1 allele) and *CYP1A1* 4889A > G (m2/m2 genotype and m2 allele), between study and control groups were statistically significant (Table 3).

The studied *ABCB1* and *CYP1A1* variants influenced the risk of MM. The presence of TT genotype of *ABCB1* 2677G > T/A polymorphism increased MM risk 2.87-fold (*p* = 0.007) (Table 3). The presence of *CYP1A1*2A* (m1) and *CYP1A1*2C* (m2) alleles decreased the risk of the disease (Table 3).

### 3.3. ABCB1 and CYP1A1 Variants as a Risk Factors of Death or MM Progression

Rare homozygotes were analyzed together with heterozygotes due to small sample size. A univariate Cox analysis revealed that patients at stage III according to ISS and without auto-HSCT had a 3.03-fold and 6.05-fold increased risk of death (Table 4). Similar findings were observed in the case of disease relapse or progression in MM patients at stage III (HR = 3.12, *p* = 0.001) and without auto-HSCT (HR = 3.07, *p* = 0.001) (Table 4). Moreover, the univariate Cox analysis showed decreased risk of death in MM patients with CT + TT genotypes of *ABCB1* 3435C > T variant (HR = 0.34, *p* = 0.04) (Table 4).

The multivariate Cox regression analysis confirmed that that patients without auto-HSCT had increased risk of death (Table 5). Moreover, it showed decreased risk of death in the group of MM patients with CT + TT genotypes of *ABCB1* 3435C > T variant – HR = 0.34, *p* = 0.04 (Table 5).

The analysis of response rate in MM showed that patients at stage III or without auto-HSCT had an increased chance of progressive disease—OR = 7.66 (*p* < 0.001) or OR = 5.59 (*p* = 0.001), respectively. The studied variants of *ABCB1* and *CYP1A1* genes did not affect the response to treatment of MM patients.

### 3.4. Correlation between Analyzed Polymorphisms and MM Clinical Features

Potential relationships between clinical data and selected genotypes were analyzed. We found, that wt/m2 + m2/m2 genotypes in comparison to wt/wt genotype of *CYP1A1* 4889A > G were associated with higher free light chain ratio (618.74 vs. 177.09, *p* < 0.001), β2-microglobulin concentration (mg/L) (7.68 vs. 5.66, *p* = 0.03) and creatinine concentration (mg/dL) (2.53 vs. 1.38, *p* = 0.003) (Table 6). Moreover the presence of *CYP1A1*2A* allele (of 6235T > C variant) was associated with higher number of platelets (K/µL) (196.4 vs. 241.1, *p* = 0.01) (Table 6). We did not observe relationships of the studied variants with high-risk chromosomal abnormalities, smoking status, exposure to carcinogens, or family history of cancer.

### 3.5. Survival of MM Patients and Studied Variants

We analyzed the association between studied genotypes and survival of MM patients. In the log rank test (Figure 3 and Figure 4), without taking into account the type of treatment, the difference in OS between CC vs. CT + TT genotypes of *ABCB1* 3435C > T was observed (Figure 3). 

It was performed a log rank analysis taking into account studied variants and the type of treatment (thalidomide vs. bortezomib vs. both—thalidomide and bortezomib). We found association of CT genotype of *ABCB1* 1236C > T with the treatment type and OS (*p* = 0.014) or PFS (*p* = 0.021) (Figure 5). Taking into account the type of treatment and OS similar results were observed in the case of GT genotype of *ABCB1* 2677G > T/A variant (*p* = 0.013) and CT genotype of *ABCB1* 3435C > T variant (*p* = 0.005). Moreover, in the analysis of treatment and PFS we found association with CT genotype of *ABCB1* 1236C > T (*p* = 0.021), CT genotype of *ABCB1* 3435C > T (*p* = 0.033), wt/wt genotype of *CYP1A1* 6235T > C (*p* = 0.016) and wt/wt genotype of *CYP1A1* 4889A > G (*p* = 0.021) (Figure 5).

### 3.6. In Vitro Study with Bortezomib

In in vitro studies, bortezomib increased the number of apoptotic and necrotic cells in all studied genotypes. The higher number of apoptotic cells was observed at 1 nM and 2 nM of bortezomib in patients with wt/wt genotype (of *CYP1A1* 4889A > G) and CC genotype (of *ABCB1* 3435 C > T) in comparison to those with wt/m2 + m2/m2 genotypes (20.67% vs. 13.06%, *p* = 0.02) and CT + TT genotypes (21.90% vs. 16.60%, *p* = 0.03), respectively (Figure 6A,B). Higher number of viable cells was found at 1 nM and 12 nM of bortezomib in cells with wt/m2 + m2/m2 (of *CYP1A1* 4889A > G) and GT + TT genotypes (of *ABCB1* 2677G > T/A) in comparison to wt/wt and GG genotypes, respectively (Figure 6C,D). In the case of 3435C > T variant was observed tendency for lower number of viable cells with CC genotype in comparison to CT + TT genotypes at 4 nM and 12 nM of bortezomib—58.8% vs. 67.65% and 48.87% vs. 54.41%, respectively (Figure 6E).

## 4. Discussion

In the current study, we analyzed the correlation of *ABCB1* and *CYP1A1* variants with the risk and the outcome of MM, as well as response to bortezomib treatment under in vitro conditions. We found relationships between the *ABCB1* 2677G > T/A, *CYP1A1* 6235T > C and 4889A > G variants with the risk of MM. Moreover, we found that the *ABCB1* 3435C > T variant affected the OS. 

The mechanisms associated with ABC transporters are responsible for the development of the MDR phenotype in MM patients [8]. The tumor cells were able to overcome the drugs cytotoxicity. This was the main cause of treatment failure in MM [8]. The identification of new prognostic and predictor factors may help to define subgroups of patients that most likely can benefit from chemotherapy [49]. Among analyzed variants of the *ABCB1* gene, including 3435C > T (synonymous variant), 1236C > T (synonymous variant) and 2677G > T/A (non-synonymous variant), the most important seems to be the one located at nucleotide 3435 [50]. It leads to changes at the mRNA level and affects protein folding [51]. Some studies have confirmed the relationship of the *ABCB1* 3435 C > T variant with MM risk. However, published data are still inconclusive [50]. The meta-analysis performed by Razi and co-workers did not find an association of *ABCB1* 3435C > T with MM susceptibility [50]. In contrast, we found the relationship of TT genotypes of 3435C > T with an almost threefold increased risk of MM development. It is possible that the *ABCB1* 3435C > T variant is in linkage disequilibrium with some alleles, which may impact MM development [6]. Similarly, the presence of the T-allele of *ABCB1* 2677G > T/A variant was associated with a higher MM risk. Drain et al. in a study of 134 MM patients, found an association between the CC genotype of *ABCB1* 3435C > T variant with shorter OS, which is in line with our observations [8]. Similar results were obtained by Drain et al. in another study [52]. Moreover, they did not find a relationship between the *ABCB1* 1236C > T and 2677G > T/A variants with OS [8]. In our study, we observed that allele frequencies and distributions of analyzed *ABCB1* variants were comparable in MM patients and healthy blood donors, which is in agreement with previous reports concerning healthy Caucasian populations [5,6]. Silent variants at the 3435 nucleotide of the *ABCB1* gene can alter ABCB1 protein conformation, which may change the specificity of binding with substrates [53]. *ABCB1* 3435C > T variant is present in the gene coding sequence (exon 26), and it is likely linked to other regions that regulate gene expression, for example the promoter or enhancer sequence or regions associated with mRNA processing [49]. It is suggested that the *ABCB1* 3435C > T variant may be significant only in individuals with specific carcinogen exposure [6]. However, in our study we did not find an association of the analyzed variants with carcinogen exposure, including smoking. Our findings suggest that the T-alleles (CT and TT genotypes) of *ABCB1* 3435C > T variant might be associated with a lower risk of death in MM patients. Univariate and multivariate Cox analysis suggested that the presence of T (at 3435 nucleotide) might be a positive prognostic factor. The T-allele of this variant can be associated with lower *ABCB1* gene expression in comparison to C-allele [54]. Moreover, we observed longer OS for patients with the CT genotype of *ABCB1* 3435C > T treated with thalidomide and bortezomib in comparison to those with CT genotype treated with thalidomide or bortezomib. This effect might be not observed in TT genotypes due to the small sample size. In the case of second analyzed synonymous variant of *ABCB1* gene—1236C > T, we found only association of CT genotype with treatment type and OS/PFS. A more detailed analysis on a larger cohort would be recommended.

The presented study showed that *CYP1A1*2A* and *CYP1A1*2C* alleles decreased the risk of MM development. Similar results were obtained by Kang et al., 2008 [30]. However, their study was conducted among an Asian population [30]. The previous report by Lincz et al. suggested no association of *CYP1A1* variants with MM in Caucasians [55]. In our study, the *CYP1A1* 4889A > G variant was not in HWE. The *CYP1A1*2A* (m1) and *CYP1A1*2C* (m2) alleles may enhance the enzyme activity and therefore the metabolism of corresponding substrates may be at a higher level [23]. *CYP1A1* 6235T > C variant raises the risk for solid tumors, for example lung cancer, cervical cancer, prostate cancer and laryngeal cancer [56,57,58,59]. In contrast, negligible relations between *CYP1A1* 6235T > C variant and gastric cancer, colorectal cancer, breast cancer and esophageal cancer risks were found [60,61,62,63]. Therefore this variant may play a diverse role in different cancers. 

The clinical stage of MM can be measured by serum markers, including β2-microglobulin, free light chain ratio, creatinine and C-reactive protein levels followed by confirmation with invasive biopsy of bone marrow [64]. In our study, we found an association of the *CYP1A1*2C* (m2) allele with a higher free light chain ratio, and higher β2-microglobulin and creatinine concentrations, which are negative prognostic factors. In contrast, the presence of the *CYP1A1*2C* allele decreased the risk of MM. These results are not consistent and require studies on a larger cohort. Moreover, *CYP1A1* 4889A > G genotypes were not in HWE, which suggest their role in disease susceptibility. However, due to low sample size this result could be obtained by error sampling.

Interactions between MM cells and the bone marrow microenvironment play a critical role in the development of MDR [65]. We observed the relationship of the *ABCB1* and *CYP1A1* variants with response to bortezomib treatment under in vitro conditions. In the case of T-allele of *ABCB1* 2677G > T/A, we noted a statistically significant higher number of viable cells at 12nM of bortezomib. However, we did not observe significant changes in the level of apoptotic and necrotic cells, taking into account the presence of the T-allele of *ABCB1* 2677G > T/A. It is possible that this result was obtained accidentally or that higher doses of bortezomib should be applied. Similarly, the presence of a T-allele of *ABCB1* 3435 C > T variant was associated with a lower number of apoptotic cells, but only at 2 nM of bortezomib. The *CYP1A1*2C* (m2) allele affected the number of viable cells and apoptotic cells only at 1nM of bortezomib. More reliable results would be associated with a significant change in the number of cells with increasing bortezomib dose. It would be recommended to repeat the *in-vitro* experiment with modified conditions, including the replacement of RPMI with AIM-V media. RMPI and 10% FCS induce apoptosis of primary B-cells [66]. 

There are some limitations of our study regarding the number of MM patients, as well as apoptosis detection in the in vitro experiment. The number of participants in the study group was relatively small, which was due to the low incidence of MM. However, the number of 110 MM patients was enough for most analyzes. Fluorescent microscopy was used to evaluate apoptosis the in in-vitro study. A more accurate method for apoptosis detection is FACS (flow cytometry-based apoptosis detection). Unfortunately, during the experiment time, FACS was not available to us. The set used for apoptosis analysis was dedicated and validated to fluorescent microscopy.

Several large scale genome-wide association studies (GWASs) have failed to identify *ABCB1* and *CYP1A1* polymorphisms in association with MM [67,68,69]. Although GWAS was made in the United States and in European ancestry populations, it is possible that in the Polish population the studied variants may have a prognostic significance [69,70,71]. In the case of *ABCB1* variants, a similar study in the Polish population was made by Jamroziak and coworkers [6]. They did not find whether common *ABCB1* variants affect predisposition to MM [6]. However, we found an association of the *ABCB1* 2677G > T/A variant with an increased risk of MM. This result might be due to error sampling or due to the fact that the study included the southwestern Polish population. Further analysis in this field would be recommended. 

## 5. Conclusions

In spite of this study’s limitations and the need for prospective studies with larger sample sizes, our findings suggest that the T-allele of *ABCB1* 2677G > T/A increased the risk of MM. In contrast, *CYP1A1*2A* (m1) and *CYP1A1*2C* (m2) alleles decreased the susceptibility of this disease in the Caucasian population in southeastern Poland. Moreover, the C-allele of *ABCB1* 3435C > T was associated with shorter OS. Further analysis on a larger cohort could help to better understand the significance of studied variants in the development and outcome of MM.

## Figures and Tables

**Figure 1 jcm-10-05276-f001:**
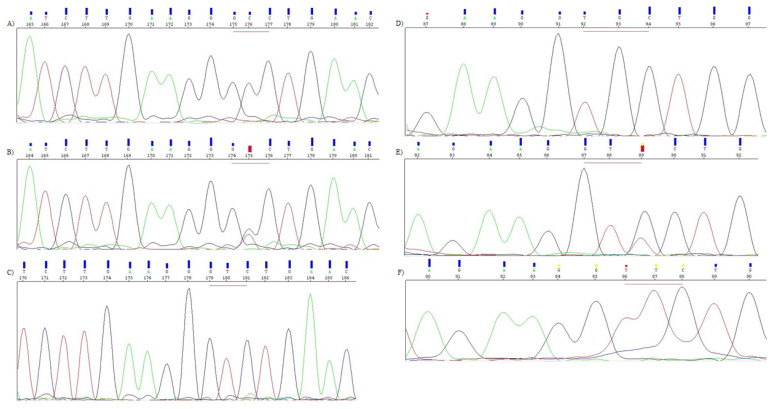
Electropherograms of *ABCB1* gene 1236C > T (**A**–**C**) and 2677G > T/A (**D**–**F**) variants obtained by automated Sanger DNA sequencing. A-CC genotype, B-CT genotype, C-TT genotype, D-GG genotype, E-GT genotype, F-TT genotype.

**Figure 2 jcm-10-05276-f002:**
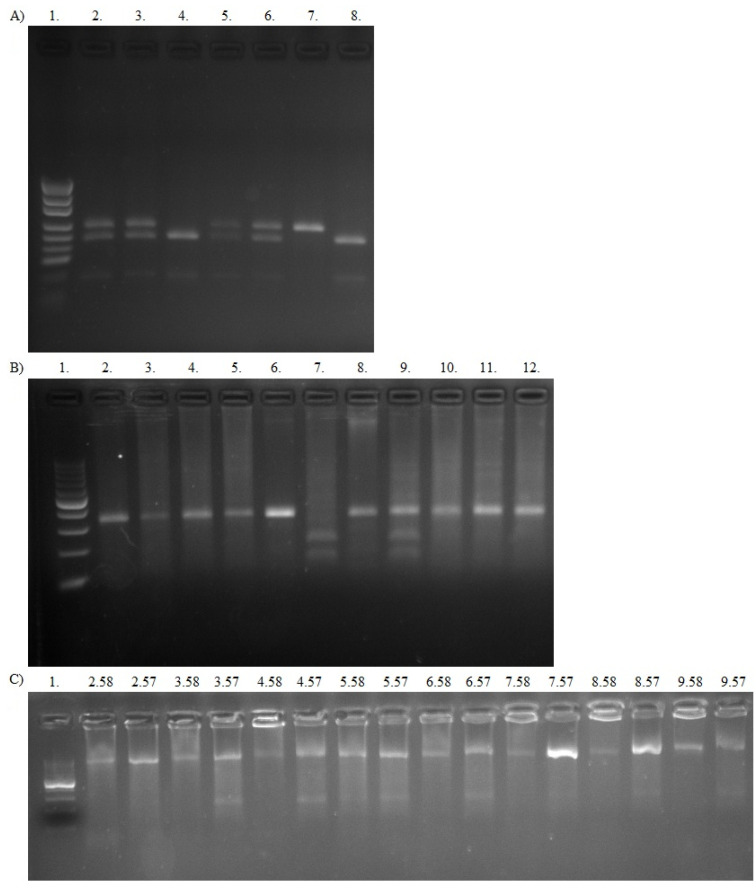
Electropherograms of studied polymorphisms obtained by PCR-RFLP method. (**A**) *ABCB1* 3435C > T SNP, Lane 1—Ladder (consists of 34, 67, 110, 147, 190, 242, 331, 404, 489 bp bands); Lanes 4 and 8 contain 172 bp and 60 bp bands (CC genotype); Lanes 2, 3, 5 and 6 contain 238 bp, 172 bp and 60 bp (CT genotype); Lane 7 contains 238 bp (TT genotype); (**B**) *CYP1A1* 6235T > C variant, Lane 1—Ladder (100 bp); Lanes 2–6, 8 and 10–12 contain 340 bp (wild type allele); Lane 9 shows 340 bp, 200 bp and 140 bp (heterozygote); Lane 7 shows 200 bp and 140 bp (variant allele, m1, *CYP1A1*2A*); (**C**) *CYP1A1* 4889A > G variant, Lane 1—Ladder (100 bp), Lanes 2–4 and 6–8 show wild type alleles; Lane 5 shows heterozygote; Lane 9 shows variant allele, m2 (*CYP1A1*2C*).

**Figure 3 jcm-10-05276-f003:**
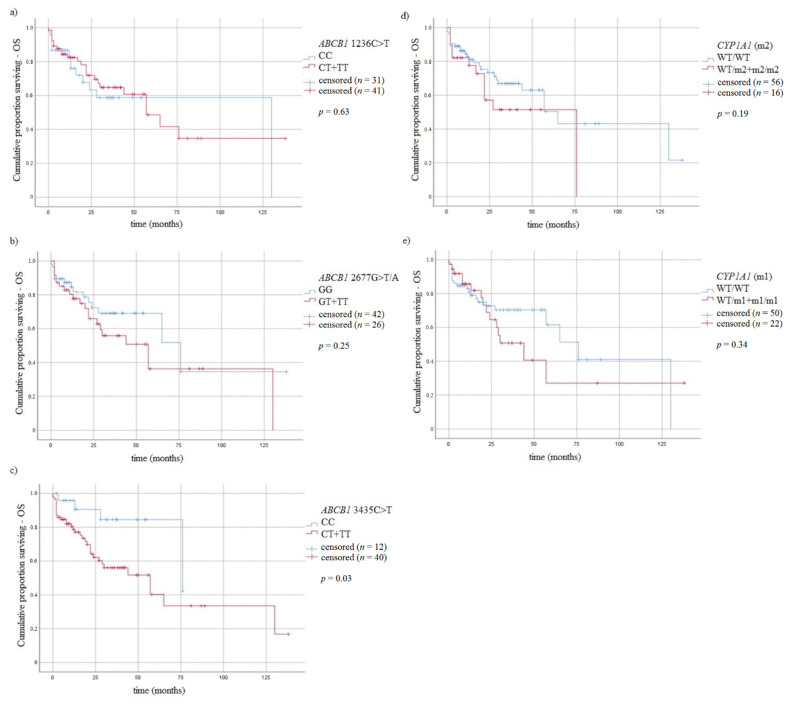
Kaplan-Meier analysis of OS in MM patients with (**a**) *ABCB1* 1236C > T genotypes, log-rank test *p* = 0.63; (**b**) *ABCB1* 2677G > T/A genotypes, log-rank test *p* = 0.25; (**c**) *ABCB1* 3435C > T, log-rank test *p* = 0.03; (**d**) *CYP1A1* 4889A > G genotypes log-rank test *p* = 0.19; (**e**) *CYP1A1* 6235T > C genotypes, log-rank test *p* = 0.34.

**Figure 4 jcm-10-05276-f004:**
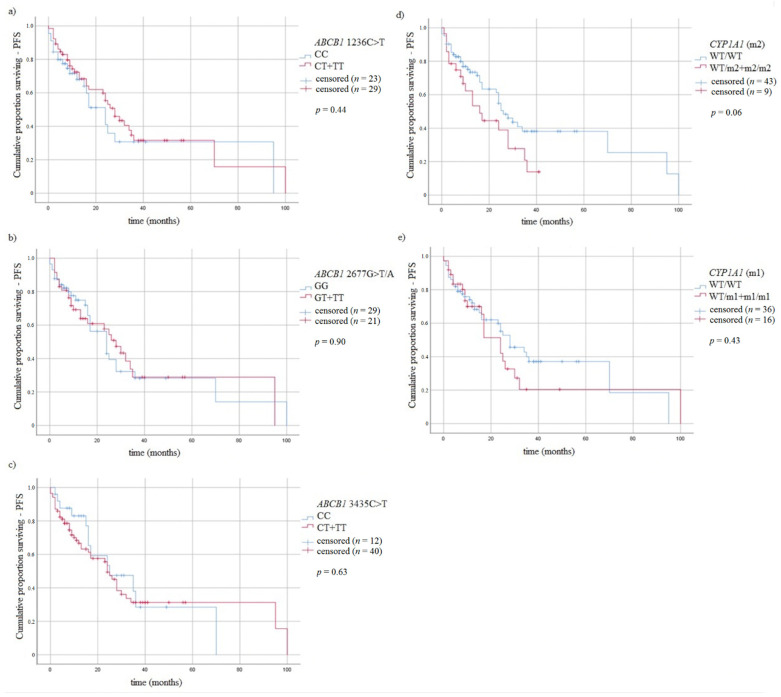
Kaplan-Meier analysis of PFS in MM patients with (**a**) *ABCB1* 1236C > T genotypes, log-rank test *p* = 0.44; (**b**) *ABCB1* 2677G > T/A genotypes, log-rank test *p* = 0.90; (**c**) *ABCB1* 3435C > T, log-rank test *p* = 0.63; (**d**) *CYP1A1* 4889A > G genotypes log-rank test *p* = 0.06; (**e**) *CYP1A1* 6235T > C genotypes, log-rank test *p* = 0.43.

**Figure 5 jcm-10-05276-f005:**
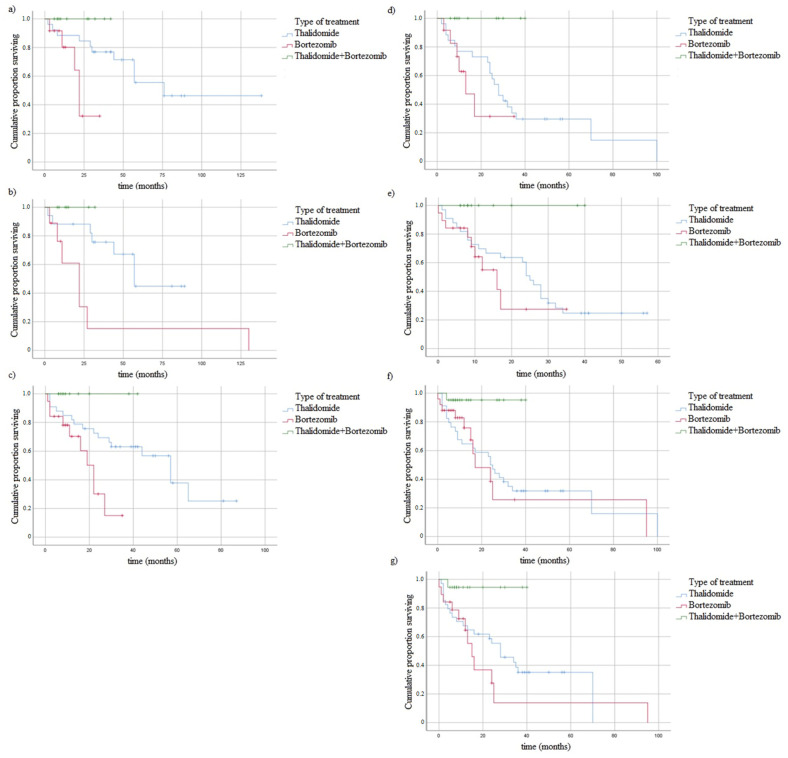
Kaplan-Meier analysis of surviving taking into account the type of treatment: (**a**) CT genotype of *ABCB1* 1236C > T (OS and treatment), log-rank test *p* = 0.014; (**b**) GT genotype of *ABCB1* 2677G > T/A (OS and treatment), log-rank test *p* = 0.013; (**c**) CT genotype of *ABCB1* 3435C > T (OS and treatment), log-rank test *p* = 0.005; (**d**) CT genotype of *ABCB1* 1236C > T (PFS and treatment), log-rank test *p* = 0.021; (**e**) CT genotype of *ABCB1* 3435C > T (PFS and treatment), log-rank test *p* = 0.033; (**f**) wt/wt genotype of *CYP1A1* 6235T > C (PFS and treatment), log-rank test *p* = 0.016; (**g**) wt/wt genotype of *CYP1A1* 4889A > G (PFS and treatment), log-rank test *p* = 0.021.

**Figure 6 jcm-10-05276-f006:**
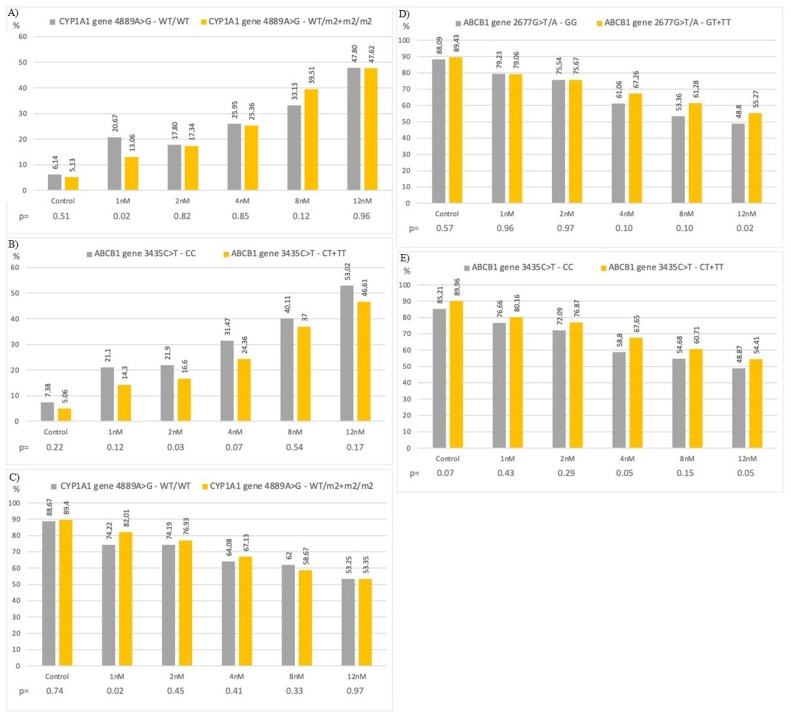
Bortezomib in vitro study. Number of: (**A**) apoptotic cells with *CYP1A1* 4889A > G genotypes; (**B**) apoptotic cells with *ABCB1* 3435C > T genotypes; (**C**) viable cells with *CYP1A1* 4889A > G genotypes; (**D**) viable cells with 2677G > T/A genotypes; (**E**) viable cells with *ABCB1* 3435C > T genotypes.

**Table 1 jcm-10-05276-t001:** The characteristics of MM patients included to the study.

Variables	MM Patients, *n* = 110
Age (years) *	
Median	65.36
Range	42–83
Sex	
Male	53 (48.18%)
Female	57 (51.82%)
Type of MM *	
IgG	60 (54.54%)
IgA	27 (24.54%)
Light chain	23 (20.9%)
Serum M protein (g/dL) *	
Median	4.96
Range	1.06–8.3
Stage according to the International Staging System *	
I	30 (27.27%)
II	33 (30%)
III	47 (42.72%)
Smoking	
Yes	20 (18.18%)
No: Non-smokers	77 (70%)
No: Ex-smokers	13 (11.81%)
Serum β2-microglobulin *	
≤3.5 mg/L	34 (30.9%)
>3.5 mg/L	33 (30%)
>5.5 mg mg/L	43 (39.1%)
Hypercalcemia *	
No	97 (88.18%)
Yes	13 (11.81%)
Renal failure *	
No	86 (78.18%)
Yes	24 (21.81%)
The stage of chronic kidney disease (grade) *	
G1	31 (28.18%)
G2	30 (27.27%)
G3A	17 (15.45%)
G3B	15 (13.63%)
G4	8 (7.27%)
G5	9 (8.18%)
Anemia grade before treatment (WHO) *	
Absent	30 (27.27%)
I—mild	36 (32.73%)
II—moderate	32 (29.09%)
III—severe	12 (10.9%)
Cytogenetic changes *	
del(17p13.1)	11 (10%)
del(17p13.1) and t(4;14)	4 (3.63%)
del(17p13.1) and t(14;16)	1 (0.9%)
t(4;14)	12 (10.9%)
t(14;16)	1 (0.9%)
Other IgH gene rearrangement	13 (11.81%)
First line treatment	
CTD	52 (47.27%)
VCD	33 (30%)
VTD	23 (20.9%)
Transplant	
ASCT	35 (31.81%)
Second line treatment	
Rd	16 (14.54%)
VCD	18 (16.36%)
VD	5 (4.54%)
Number of relapses	
1	31 (28.18%)
2	8 (7.27%)
Follow-up (months)	
Median	18.5
Range	2–138
Deaths	
Before chemotherapy	2 (1.81%)
Total	35 (31.81%)

* at diagnosis, M-mean, SD—standard deviation. Abbreviations: ASCT—autologous stem cell transplant; Rd—Lenalidomide, Dexamethasone; CTD—Cyclophosphamide, Thalidomide, Dexamethasone; VCD—Bortezomib, Cyclophosphamide, Dexamethasone; VTD—Bortezomib, Thalidomide, Dexamethasone.

**Table 2 jcm-10-05276-t002:** Hardy-Weinberg equilibrium (HWE) for *ABCB1* and *CYP1A1* variants in case and control groups according to expected (E) and observed (O) values.

Groups	Genotypes			HWE *p* Value and χ^2^ *
*ABCB1* gene 1236C > T				
-	CC	CT	TT	-
CONTROL				
E	32.49	49.02	18.49	*p* = 0.46, χ^2^ = 0.54
O	30	54	16	
CASE				
E	43.91	51.17	14.91	*p* = 0.72 χ^2^ = 0.12
O	45	49	16	
*ABCB1* gene 2677G > T/A				
-	GG	GT + GA	AA + TT + AT	-
CONTROL				
E	29.16	49.68	21.16	*p* = 0.16, χ^2^ = 1.93
O	34	40	26	
CASE				
E	51.82	47.36	10.82	*p* = 0.08, χ^2^ = 3.01
O	57	37	16	
*ABCB1* gene 3435C > T				
-	CC	CT	TT	-
CONTROL				
E	19.36	49.28	31.36	*p* = 0.17, χ^2^ = 0.67
O	18	52	30	
CASE				
E	30	54.90	25.10	*p* = 0.17, χ^2^ = 1.88
O	25	65	20	
*CYP1A1* gene 6235T > C (m1, *CYP1A1*2A*)				
-	WT	WT/m1	m1/m1	-
CONTROL				
E	79.21	19.58	1.21	*p* = 0.07, χ^2^ = 3.27
O	82	14	4	
CASE				
E	70.4	35.2	4.4	*p* = 0.64, χ^2^ = 0.42
O	72	32	6	
*CYP1A1* gene 4889A > G (m2, *CYP1A1*2C*)				
-	WT	WT/m2	m2/m2	-
CONTROL				
E	90.25	9.5	0.25	*p* = 0.55, χ^2^ = 0.35
O	90	10	0	
CASE				
E	76.94	30.1	2.94	*p* = 0.002, χ^2^ = 9.07
O	82	20	8	

* if the χ^2^ ≤ 3.84 and the corresponding *p* ≥ 0.05 then the population is in HWE.

**Table 3 jcm-10-05276-t003:** The comparison of allele frequency and distribution of *ABCB1* and *CYP1A1* variants among MM patients and controls.

Gene Variants and Alleles	MM *n* (%)	Controls *n* (%)	Odds Ratio	95% CI	*p* Values
*ABCB1* gene 1236C > T					
CC	45 (41%)	30 (30%)	1	-	-
CT	49 (44.5%)	54 (54%)	1.65	0.90–3.02	0.10
TT	16 (14.5%)	16 (16%)	1.50	0.65–3.44	0.33
Total:	110 (100%)	100 (100%)			
C	139 (63%)	114 (57%)	1	-	-
T	81 (37%)	86 (43%)	1.29	0.87–1.91	0.19
Total:	220 (100%)	200 (100%)			
*ABCB1* gene 2677G > T/A					
GG	57 (51.7%)	34 (34%)	1	-	-
GT	33 (30%)	36 (36%)	1.82	0.97–3.45	0.06
TT	14 (12.8%)	24 (24%)	2.87	1.31–6.29	0.007
GA	4 (3.7%)	4 (4%)	1.67	0.39–7.14	0.74
AT	1 (0.9%)	2 (2%)	3.35	0.29–38.38	0.67
AA	1 (0.9%)	0	*	*	*
Total:	110 (100)	100 (100)			
G	152 (69%)	108 (54%)	1	-	-
T	62 (28%)	86 (43%)	1.95	1.29–2.93	0.001
A	6 (3%)	6 (3%)	1.40	0.44–4.48	0.55
Total:	220 (100)	200 (100)			
*ABCB1* gene 3435C > T					
CC	25 (22.7%)	18 (18%)	1	-	-
CT	65 (59.1%)	52 (52%)	1.11	0.54–2.25	0.76
TT	20 (18.2%)	30 (30%)	2.08	0.90–4.77	0.08
Total:	110 (100)	100 (100)			
C	114 (52%)	88 (44%)	1	-	-
T	106 (48%)	112 (56%)	1.36	0.93–2.01	0.10
Total:	220 (100)	200 (100)			
*CYP1A1* gene 6235T > C (m1, *CYP1A1*2A*)					
WT/WT	72 (65.5%)	82 (82%)	1	-	-
WT/m1	32 (29.1%)	14 (14%)	0.38	0.19–0.77	0.006
m1/m1	6 (5.4%)	4 (4%)	0.68	0.15–2.15	0.62
Total:	110 (100)	100 (100)			
WT	176 (80%)	178 (89%)	1	-	-
m1	44 (20%)	22 (11%)	0.49	0.28–0.85	0.011
Total:	220 (100)	200 (100)			
*CYP1A1* gene 4889A > G (m2, *CYP1A1*2C*)					
WT/WT	82 (74.5%)	90 (90%)	1	-	-
WT/m2	20 (18.2%)	10 (10%)	0.45	0.20–1.03	0.054
m2/m2	8 (7.3%)	0	*	*	0.011
Total:	110 (100)	100 (100)			
WT	185 (84%)	190 (95%)	1	-	-
m2	35 (16%)	10 (5%)	0.27	0.13–0.57	0.0003
Total:	220 (100)	200 (100)			

* too small group for analysis.

**Table 4 jcm-10-05276-t004:** Univariate Cox analysis in survival of MM patients.

Variable	Univariate Cox Analysis for OS			Univariate Cox Analysis for PFS		
	*p* Value	HR	95% CI	*p* Value	HR	95% CI
ISS						
I + II	-	R	-	-	R	-
III	0.001	3.03	0.17–0.64	0.001	3.12	0.18–0.54
Auto-HSCT						
yes	-	R	-	-	R	-
no	<0.001	6.05	2.44–14.97	0.001	3.07	1.68–5.62
*ABCB1* gene 1236C > T						
CC	-	R	-	-	R	-
CT + TT	0.64	1.17	0.60–2.30	0.45	1.23	0.72–2.10
*ABCB1* gene 2677G > T/A *						
GG	-	R	-	-	R	-
GT + TT	0.26	0.68	0.35–1.33	0.91	1.03	0.60–1.76
*ABCB1* gene 3435C > T						
CC	-	R	-	-	R	-
CT + TT	0.04	0.34	0.12–0.96	0.64	0.86	0.46–1.60
*CYP1A1* gene 6235T > C (m1, *CYP1A1*2A*)						
WT/WT	-	R	-	-	R	-
WT/m1 + m1/m1	0.35	0.73	0.38–1.41	0.44	0.80	0.46–1.39
*CYP1A1* gene 4889A > G (m2, *CYP1A1*2C*)						
WT/WT	-	R	-	-	R	-
WT/m2 + m2/m2	0.20	0.64	0.32–1.27	0.07	0.60	0.34–1.05

R-reference; * A alleles were excluded from analysis due to low frequency (<5%).

**Table 5 jcm-10-05276-t005:** Multivariate Cox analysis in survival of MM patients.

Variable	Multivariate Cox Analysis for OS			Multivariate Cox Analysis for PFS		
	*p* Value	HR	95% CI	*p* Value	HR	95% CI
ISS						
I + II	-	reference	-	-	reference	-
III	0.05	0.44	0.20–0.99	0.003	0.35	0.18–0.70
Auto-HSCT						
yes	-	reference	-	-	reference	-
no	0.02	3.64	1.27–10.50	0.35	1.43	0.67–3.04
*ABCB1* gene 1236C > T						
CC	-	R	-	-	R	-
CT + TT	0.74	0.86	0.37–2.04	0.95	1.02	0.50–2.05
*ABCB1* gene 2677G > T/A *						
GG	-	R	-	-	R	-
GT + TT	0.56	1.26	0.56–2.83	0.33	1.36	0.73–2.53
*ABCB1* gene 3435C > T						
CC	-	R	-	-	R	-
CT + TT	0.04	0.29	0.09–0.93	0.31	0.69	0.34–1.40
*CYP1A1* gene 6235T > C (m1, *CYP1A1*2A*)						
WT/WT	-	R	-	-	R	-
WT/m1 + m1/m1	0.17	0.60	0.29–1.23	0.11	0.61	0.33–1.13
*CYP1A1* gene 4889A > G (m2, *CYP1A1*2C*)						
WT/WT	-	R	-	-	R	-
WT/m2 + m2/m2	0.51	0.76	0.33–1.73	0.29	0.69	0.35–1.37

* A alleles were excluded from analysis due to low frequency (<5%).

**Table 6 jcm-10-05276-t006:** The clinical values of MM patients included to the study taking into account studied polymorphisms.

Variables	MM Patients	1236C > T	2677G > T/A	3435C > T	6235T > C (m1, *CYP1A1*2A*)	4889A > G (m2, *CYP1A1*2C*)
		CC vs. CT + TT *p*-Value	GG vs. GT + TT *p*-Value	CC vs. CT + TT *p*-Value	WT/WT vs. WT/m1 and m1/m1 *p*-Value	WT/WT vs. WT/m and m2/m2 *p*-Value
Mean age (years) *	65.36	0.11	0.67	0.69	0.70	0.26
Free light chain ratio *	292.64	0.15	0.21	0.17	0.89	<0.001
% of plasma cells in bone marrow *	30.85	0.16	0.10	0.78	0.24	0.37
Albumins (g/dL) *	3.58	0.88	0.57	0.76	0.71	0.31
β2-microglobulin * (mg/L)	6.18	0.53	0.18	0.71	0.63	0.03
Calcium * (mM/L)	2.43	0.22	0.83	0.37	0.19	0.64
Hemoglobin * (g/dL)	10.58	0.89	0.62	0.45	0.94	0.31
Creatinine * (mg/dL)	1.66	0.44	0.61	0.77	0.44	<0.001
Platelets (K/μL)	210.64	0.54	0.83	0.32	0.01	0.26
C-reactive protein * (mg/L)	15.16	0.96	0.60	0.63	0.88	0.08
Estimated glomerular filtration rate * mL/min/1.73 m^2^	60.31	0.50	0.56	0.09	0.20	0.87

* at diagnosis.

## Data Availability

The clinical data used to support the findings of this study are available from the corresponding author upon request.

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
