# Peer review of "The Relationship of ABCB1/MDR1 and CYP1A1 Variants with the Risk of Disease Development and Shortening of Overall Survival in Patients with Multiple Myeloma"

_jcm, 2021, doi:10.3390/jcm10225276_

Round 1

Reviewer 1 Report

Multiple myeloma is the second most common hematological cancer, and it is mostly incurable despite the efforts to find new treatments. The authors of the paper are suggesting a new way to classify patients according to the variants they bear ABCB1 and CYP1A1 and especially the response to bortezomib treatment .

Here are my comments:

1) At this concern, I found the introduction lacking some clarifications about the reasons the authors choose these two genes especially and the link with bortezomib metabolism.

2) I was wondering if the use of healthy controls with a mean of age of 34 years is adequate, given the mean age of MM patients is 65 years and that MM is mostly developed by aged persons. Authors should justify the comparison with young healthy donors concerning the conclusion on the enhanced risk of MM with the indicated genotype.

3) It seems to me that the survival is better with the combination of Thalidomide and bortezomib treatment, however in the discussion the authors mentioned that they observed longer OS for patients with CT genotype of ABCB1 3435C>T treated with bortezomib in comparison with the other treatments or am i missing something? 

4) If I correctly interpreted the Figure 6, I conclude that the treatment with the Thalidomide combined to Bortezomib enhances the OS and PFS of MM patients independently of the genotypes. Am I right? I suggest otherwise to compare the OS and PFS  between the patients with ABCB1 3435 CT+TT vs CC (as in figure 4-5) in relation with the treatment (bortezomib, Thalidomide or combined).

5) By experience, the viability of plasmocytes is weak in vitro and it is impossible to maintain these cells in culture. As the authors mentioned in the text, FACS analysis is more reliable for this kind of results. The FACS is more accurate because it allows us to differentiate easily between all types of cells and well gate the malignant plasmocytes by their specific markers. It would be interesting to show some fluorescence microscopy images showing the differences between plasmocytes and other cell types at the size level in addition to one condition where bortezomib showed good results. 

6) The results in Table 9 are very hard to read and so to understand. This kind of experiments is better represented by graphs, would it be possible to change this table with graphs in order to make it easily comprehensive by the reader? It will also be useful to put more details in the legend. 

7) For the reference 42, I think the authors made a mistake and the right reference is https://doi.org/10.3389/fonc.2019.00044

"ACE Insertion/Deletion Polymorphism (rs4646994) Is Associated With the Increased Risk of Multiple Myeloma"

8) On page 16, line 318, There is one error. The author wrote 1235 instead of 1236

Author Response

Please see the attachment with reply to your review report.

Reviewer 2 Report

In this manuscript, Zmorzynski et al. have investigated frequency and clinical impact of ABCB1 and CYP1A1 polymorphisms in multiple myeloma (MM) outcome. These results might suggest candidate pharmacogenomics biomarkers of responsiveness to therapy and clinical outcome in MM; however, some questions should be addressed.

  1. ABCB1 is not the only mechanism that might be responsible of drug resistance in MM, as it also depends on other biological features of the disease and drugs used. In the introduction, the biological link between ABCB1 and CYP1A1 is not well presented.
  2. There are too many tables and figures. Please consider combining some (e.g., figures 1 and 2) and/or moving something to supplemental materials and methods, including some better descriptions of experimental design. Indeed, some assays are very well described, while others are lacking several information (e.g., 2.6 bortezomib in vitro treatment).
  3. Table 1 is missing information, such as range for age, β2-microglobulin levels, serum calcium levels, monoclonal component levels, cytogenetic abnormalities other than del17p and involving the IgH gene (e.g., del13q), number of relapses, number of patients who received a hematopoietic stem cell transplantation (and type), second-line treatment and number of patients who received it, number of total deaths, median follow-up. Abbreviations should be stated in table’s legend and used as they are in the table (e.g., CTD, VTD,…). Exposure to carcinogenic factors and family history of cancers can be removed as 1) this Reviewer is not aware of substances that are causing MM or are related to its development, and 2) these statements are not precise. Table 2 can be removed.
  4. Results should be organized in subsections and more detailed. Consider consolidate or remove some tables and make them easier to read removing not necessary data (e.g., in table 3, the “total” column and/or the chi square value). With all these tables, the reader got lost and does not easily take the message.
  5. On lines 269-271 and lines 278-279, the Authors stated that without auto-HSCT, patients have a worse prognosis, that is a very established phenomenon. Therefore, what is the association between this statement and studied variants’ frequency? Patients with late-stage disease were more likely to have the risk variant? Moreover, the Authors should not stress too much results with p values between 0.01 and 0.05 because the number of patients is small (110) and for analysis of polymorphism a much higher number of cases should be presented in order to draw conclusions. Indeed, it would be good if the Authors confirm their initial findings on larger cohorts using available online databases (e.g., NCBI GEO database).
  6. By definition, stable disease is when it’s not meeting criteria for CR, VGPR, PR or progressive disease. Please discuss why SD was included within responders.
  7. In table 5 and 6, what do “yes” and “no” stay for?
  8. In table 8, p values are sufficient otherwise the table is too crowded.
  9. Mantel-Cox log rank test should be used for comparison of OS and PFS data for all conditions (and not ANOVA). Please reorganized Figures 4 and 5 showing as labels number of censored subjects and p values. The Authors should try to combine all ABCB1 variants in one survival graph, as well as for CYP1A1, and thus showing in one figure OS and PFS.
  10. The Authors should consider to show results on apoptosis using graph bars instead of tables. Representative dot plots should be displayed.

Author Response

Dear Reviewer,

thank you for all suggestions and substantive comments to the manuscript. We have taken them into account in the revised version of the manuscript. Below are listed detailed answers to your review report.

In this manuscript, Zmorzynski et al. have investigated frequency and clinical impact of ABCB1 and CYP1A1 polymorphisms in multiple myeloma (MM) outcome. These results might suggest candidate pharmacogenomics biomarkers of responsiveness to therapy and clinical outcome in MM; however, some questions should be addressed.

  1. ABCB1 is not the only mechanism that might be responsible of drug resistance in MM, as it also depends on other biological features of the disease and drugs used. In the introduction, the biological link between ABCB1 and CYP1A1 is not well presented.

Response:

-Done as suggested. In the introduction additional information about the link of ABCB1 and CYP1A1 genes, as well as their relationship with bortezomib metabolism were added.

  1. There are too many tables and figures. Please consider combining some (e.g., figures 1 and 2) and/or moving something to supplemental materials and methods, including some better descriptions of experimental design. Indeed, some assays are very well described, while others are lacking several information (e.g., 2.6 bortezomib in vitro treatment).

Response:

-Done as suggested. Figures 1 and 2 were combined. Table 9 was removed and replaced by Figure 6 (with graphs). The in vitro bortezomib experiment was better described. However, cytogenetic analysis and bortezomib in vitro treatment were shortly described because these parts were described previously. In the case of cytogenetics the detailed description was published by A. DmoszyÅ„ska and S. Chocholska, “Molecular Biology Methods in the Diagnosis of Multiple Myeloma,” 2012, pp. 443–449.  Bortezomib analysis was described in paper - S. Zmorzynski et al., “ACE Insertion/Deletion Polymorphism (rs4646994) Is Associated With the Increased Risk of Multiple Myeloma,” Front. Oncol., vol. 9, Feb. 2019, doi: 10.3389/fonc.2019.00044.

  1. Table 1 is missing information, such as range for age, β2-microglobulin levels, serum calcium levels, monoclonal component levels, cytogenetic abnormalities other than del17p and involving the IgH gene (e.g., del13q), number of relapses, number of patients who received a hematopoietic stem cell transplantation (and type), second-line treatment and number of patients who received it, number of total deaths, median follow-up. Abbreviations should be stated in table’s legend and used as they are in the table (e.g., CTD, VTD,…). Exposure to carcinogenic factors and family history of cancers can be removed as 1) this Reviewer is not aware of substances that are causing MM or are related to its development, and 2) these statements are not precise. Table 2 can be removed.

Response:

- Done as suggested.

  1. Results should be organized in subsections and more detailed. Consider consolidate or remove some tables and make them easier to read removing not necessary data (e.g., in table 3, the “total” column and/or the chi square value). With all these tables, the reader got lost and does not easily take the message.

Response:

- Done as suggested. Table 3 is now table 2.

Table 7 (entiteled - ABCB1 and CYP1A1 variants in response rate of MM patients) was removed.

  1. On lines 269-271 and lines 278-279, the Authors stated that without auto-HSCT, patients have a worse prognosis, that is a very established phenomenon. Therefore, what is the association between this statement and studied variants’ frequency? Patients with late-stage disease were more likely to have the risk variant? Moreover, the Authors should not stress too much results with p values between 0.01 and 0.05 because the number of patients is small (110) and for analysis of polymorphism a much higher number of cases should be presented in order to draw conclusions. Indeed, it would be good if the Authors confirm their initial findings on larger cohorts using available online databases (e.g., NCBI GEO database).

Response:

-in our analysis, we did not find the association between studied variants and presence/absence of auto-HSCT. Moreover, we did not find relationship of studied variants with the stage of diseases. Multiple myeloma is a rare hematological disorder and we tried to show our results in the context of many clinical data, as well as OS and PFS. In the future, we will prepare mata-analysis and we will take into account our findings (from our previous studies too) on larger cohorts using available online databases.    

  1. By definition, stable disease is when it’s not meeting criteria for CR, VGPR, PR or progressive disease. Please discuss why SD was included within responders.

Response:

In a cohort study, only one stable disease (SD) was reported. Patient remained on VD treatment for 7 months with a clinical benefit. Patient was not eligible for ASCT and intensive chemotherapy. SD was considered as a satisfactory response in this case.

  1. In table 5 and 6, what do “yes” and “no” stay for?

Response:

-the terms “yes” or “no” refer to patients who have undergone auto-HSCT (=yes) or not (=no). The number of tables has been changed. Therefore, Table 5 is now Table 4, and Table 6 is now Table 5.

  1. In table 8, p values are sufficient otherwise the table is too crowded.

Response:

-table 8 is now table 6. Only p-values were left in table.

  1. Mantel-Cox log rank test should be used for comparison of OS and PFS data for all conditions (and not ANOVA). Please reorganized Figures 4 and 5 showing as labels number of censored subjects and p values. The Authors should try to combine all ABCB1 variants in one survival graph, as well as for CYP1A1, and thus showing in one figure OS and PFS.

Response:

-you are right, that for OS/PFS comparison Mantel-Cox rank test should be applied. He have removed results (for OS/PFS) obtained by ANOVA. Figures 4 and 5 (now 3 and 4) were reorganized and numbers of censored subjects and p-values were shown in the diagrams. We left separate charts for OS and PFS to make them more readable. Graphs plotted for both OS and PFS values, as well as for each genotype were unreadable.

  1. The Authors should consider to show results on apoptosis using graph bars instead of tables. Representative dot plots should be displayed.

Response:

- Done as suggested. Instead of dot plots, bar graphs have been shown (Figure 6).

Round 2

Reviewer 2 Report

The Authors have addressed most of the comments.

However, some minor changes are needed.

  1. Figures 3-5 should be provided in a higher resolution. Is it possible to combine ABCB1 or CYP1A1 genotype curves within one survival or PFS graph based on gene? It would be easier to compare everything at once.
  2. Figure 6. Statistical symbols should be added for significant comparisons. Please check the figure for consistency.

Author Response

REVIEWER 2

1) Figures 3-5 should be provided in a higher resolution. Is it possible to combine ABCB1 or CYP1A1 genotype curves within one survival or PFS graph based on gene? It would be easier to compare everything at once.

-RESPONSE:

Figures 3-5 were provided in a higher resolution. The figure resolution seems to be good in the file (png or jpg – we have check both types). However, after pasting into a word file the resolution of figures decreased. Certainly the photos meet the requirements of the editorial office.

It is not possible to combine ABCB1/CYP1A1 curves within one OS/PFS graph based on a gene. The statistical program does not allow to select several variables and present them on one graph.

2) Figure 6. Statistical symbols should be added for significant comparisons. Please check the figure for consistency.

-RESPONSE:

Significant values are below 0.05. The p values are added below the graphs, and there is no need to add statistical symbols like asterisks (*).

The figure 6 is correct, p values refer to difference between genotypes.